# Estrogen Regulates the Expression and Localization of YAP in the Uterus of Mice

**DOI:** 10.3390/ijms23179772

**Published:** 2022-08-29

**Authors:** Sohyeon Moon, Ok-Hee Lee, Byeongseok Kim, Jinju Park, Semi Hwang, Siyoung Lee, Giwan Lee, Hyukjung Kim, Hyuk Song, Kwonho Hong, Jaejin Cho, Youngsok Choi

**Affiliations:** 1Department of Stem Cell and Regenerative Biotechnology, Konkuk University, Seoul 05029, Korea; 2Department of Dental Regenerative Biotechnology, Seoul National University, Seoul 03080, Korea; 3Dental Research Institute, School of Dentistry, Seoul National University, Seoul 03080, Korea; 4Institute of Advanced Regenerative Biotechnology, Konkuk University, Seoul 05029, Korea

**Keywords:** YAP, estrogen, mouse uterus, estrous cycle

## Abstract

The dynamics of uterine endometrium is important for successful establishment and maintenance of embryonic implantation and development, along with extensive cell differentiation and proliferation. The tissue event is precisely and complicatedly regulated as several signaling pathways are involved including two main hormones, estrogen and progesterone signaling. We previously showed a novel signaling molecule, Serine/threonine protein kinase 3/4 (STK3/4), which is responded to hormone in the mouse uterine epithelium. However, the role and regulation of its target, YES-associated protein (YAP) remains unknown. In this study, we investigated the expression and regulation of YAP in mouse endometrium. We found that YAP was periodically expressed in the endometrium during the estrous cycle. Furthermore, periodic expression of YAP was shown to be related to the pathway under hormone treatment. Interestingly, estrogen was shown to positively modulate YAP via endometrial epithelial receptors. In addition, the knockdown of YAP showed that YAP regulated various target genes in endometrial cells. The knockdown of YAP down-regulated numerous targets including *ADAMTS1*, *AMOT*, *AMOTL1*, *ANKRD1*, *CTNNA1*, *MCL1*. On the other hand, the expressions of *AREG* and *AXL* were increased by its knockdown. These findings imply that YAP responds via Hippo signaling under various intrauterine signals and is considered to play a role in the expression of factors important for uterine endometrium dynamic regulation.

## 1. Introduction

Orchestrated regulation of uterine endometrium is crucial for successful establishment and maintenance of embryonic implantation and development [1,2]. Embryo implantation is mediated by physical signals between the mother’s uterus and trophoblasts and chemical signals, such as steroid hormones [3]. Two steroid hormones, estrogen and progesterone, are important for integrating uterine events for early pregnancy. These hormones regulate changes in the endometrial epithelium, enabling blastocysts to implant into the endometrium [4]. A series of their interaction with other signaling factors such as Wnt and hedgehog facilitates embryonic growth during early pregnancy and induces endometrium dynamic [1].

The Hippo signaling was initially reported to be involved in regulating organ size [5]. A series of phosphorylation starting with STK3/4 turned off the signal by phosphorylating the final target, YAP. Phosphorylated YAP shuttles from the nucleus to the cytoplasm. Under normal circumstances, YAP exists in the nucleus and acts as a transcriptional factor regulating many target gene expression. Connective tissue growth factor (CTGF) and cysteine-rich angiogenesis inducer 61 (CYR61) are well-characterized factors that regulate various cellular processes such as proliferation and migration [6,7,8,9]. YAP is also involved in gene expression of numerous other factors including ADAM metallopeptidase with thrombospondin type 1 motif 1 (*ADAMTS1*) [10,11], angiomotin (*AMOT*) [2,12], angiomotin like 1 (*AMOTL1*) [13], ankyrin repeat domain 1 (*ANKRD1*) [14,15], catenin alpha 1 (*CTNNA1*), MCL1 apoptosis regulator, BCL2 family member (*MCL1*) [16], amphiregulin (*AREG*), and AXL receptor tyrosine kinase (*AXL*) [17].

Recent studies have reported that the Hippo signaling pathway is regulated in reproductive system including the uterus [2]. We also reported that STK3/4 is regulated by estrogen and its receptor in the endometrium during the estrous cycle. These findings imply that the Hippo signaling system affects uterine dynamic by regulating YAP and its targets [18]. However, the role and regulatory mechanism of YAP in the uterus remains unknown. Therefore, we examined the regulation of YAP expression and its regulation in uterine endometrium.

## 2. Results

### 2.1. YAP Expression in the Mouse Uterus at Each Stage of the Estrous Cycle

First of all, we looked into the expression of *YAP* in the uterus during the estrous cycle using RT-PCR and qRT-PCR. As shown in Figure 1A,B, *YAP* transcript was dynamically expressed according to the stage of the estrous cycle. Its expression was significantly increased at the estrus, and drastically decreased to a minimum at the metestrus stage. Interestingly, its expression was elevated again in the diestrus. Evaluation of each stage of the estrous cycle was confirmed by vaginal smear analysis (Figure 1C). In smears from mice in proestrous stage, most cells consist of nucleated epithelial cells (NE), whereas smears from mice in estrus consist mostly of cornified epithelial cells (CE). In contrast, leukocytes (LK) and CE were abundantly present in smears at the metestus stage, whereas smears from diestrous stage mice contained mainly of LK.

### 2.2. YAP Localization in Mouse Uterus during the Estrous Cycle

Next, we examined the expression of YAP protein and phosphorylated YAP during the estrous cycle at the protein level. Western blot analysis confirmed that YAP and phosphorylated YAP (Ser127) were dynamically regulated during the estrous cycle. Similar to mRNA levels, both YAP and phospho-YAP (YAP S127) were significantly downregulated during the metestrus stage (Figure 2A–C). Immunofluorescence staining showed that YAP and phospho-YAP were relatively higher in endometrial glandular epithelial (GE) cells and luminal epithelial (LE) cells compared to stromal (ST) (Figure 2D). Phosphorylation of YAP results in its translocation from the nucleus to the cytoplasm, where it remains inactive; by contrast, unphosphorylated YAP is activated and translocated into the nucleus to regulate transcriptional activity [2,19,20].

### 2.3. Regulation of YAP Expression by Estrogen in the Mouse Uterus

We previously showed that serine/threonine kinase 3/4 (STK3/4) is regulated by estrogen and its signaling pathway [18]. Therefore, we next investigated whether YAP was also regulated by estrogen. We used ovariectomized (OVX) mouse model to eliminate endogenous steroid hormone mechanisms and investigated the effects of estrogen alone on YAP expression. Western blot analysis revealed that YAP protein expression was increased with time after estrogen treatment (Figure 3A). But the quantification analysis showed that there is no significant difference over time (Figure 3B). By contrast, phospho-YAP (YAP S127) levels showed a time-dependent increase after estrogen injection (Figure 3A,C). As shown in Figure 3D, immunofluorescence staining clearly demonstrated that YAP localized only to GE and LE cell nuclei before estrogen treatment (0 h) and then localized in the cytoplasm and nucleus after estrogen treatment. Furthermore, we demonstrated a time-dependent increase in phospho-YAP levels in the cytoplasm of GE and LE cells after estrogen treatment, similar to Western blot analysis.

### 2.4. Effect of the Estrogen Receptor Antagonist ICI on the Expression of YAP in OVX Mouse Uterus

We then evaluated whether the increase in YAP phosphorylation were mediated by the signaling of estrogen receptor. OVX mice were injected with the estrogen receptor antagonist, ICI 182,780 (ICI), 30 min prior to estrogen treatment to block the estrogen response through the estrogen receptor. We found that YAP was present in both cytoplasm and nucleus of LE and GE cells. Furthermore, YAP phosphorylation was also increased in the cytoplasm after estrogen treatment. However, the response by estrogen was significantly inhibited with ICI treatment (Figure 4).

### 2.5. Knockdown of YAP in Human Endometrial Epithelial Cell Lines

Because the expression and localization of YAP were regulated in endometrial epithelium, we looked into the effect of small interfering RNA-mediated knockdown of YAP on target genes in human endometrial cell line, Ishikawa cells. YAP knockdown using siYAP was confirmed by Western blot analysis after siRNA treatment (Figure 5A). YAP and phospho-YAP protein levels were significantly reduced by siYAP treatment. Next, we evaluated the expression levels of target factors known to be expressed in reproductive organs including *ADAMTS1* [10,11], *AMOT* [2,12], *AMOTL1* [13], *ANKRD1* [14,15], *CTNNA1*, *MCL1* [16], *AREG*, and *AXL* [17]. qRT-PCR analysis revealed that YAP knockdown decreased the expression of *ADAMTS1*, *AMOT*, *AMOTL1*, *ANKRD1*, *CTNNA1*, and *MCL1* but significantly increased the expression of *AREG* and *AXL* (Figure 5B).

## 3. Discussion

In this study, we investigated the regulation of YAP, a downstream target of the Hippo signaling pathway, in uterine endometrium. We found that YAP was periodically expressed in the endometrium during the estrous cycle. Moreover, we showed that periodic expression of YAP was related to a steroid hormone, estrogen. Interestingly, we found that estrogen inactivated YAP via its phosphorylation via estrogen receptor-dependent signaling.

YAP, a transcriptional activation cofactor, regulates cell proliferation and apoptosis and is associated with tumorigenesis [21]. YAP maintains spatiotemporal control of cellular behaviors through regulation of cell-environment interactions and allows for cell- and event-specific responses [22]. Conversely, YAP has dual roles in inducing apoptosis and inhibiting tumor growth. In breast cancer, the *YAP* gene is deleted, and YAP knockdown in breast cancer cells inhibits apoptosis and promotes tumor growth, suggesting that YAP can act as a tumor suppressor [23].

Therefore, YAP has variable targets depending on tissue types. We could think of a putative role for YAP in the endometrium from many related studies, including our findings. Among those, BIRC5 (called Survivin) was increased in human hepatic malignancy [24]. Interestingly, we previously demonstrated that the expression of *BIRC5* is regulated by estrogen in the uterus [25]. YAP knockdown reduced the expression of *AMOT* and *AMOTL1*. They are known as angiogenesis-related factors and have reported to be regulated by estrogen in uterine luminal epithelium and progesterone in the endometrial matrix [12,13]. In fact, the regulation of angiogenesis in uterine endometrium is crucial for maintenance of normal uterine environment [26]. CTNNA1 is known as adherens junctions with CTNNB1. Both CTNNA1 and CTNNB1 were mainly detected in luminal epithelium of the uterus [27]. Aberrant expression of CTNNB1 is found in uterine fibroids [28], endometrial hyperplasia, and endometrial cancer. This means that proper regulation of CTNNA1 and CTNNB1 is important for maintaining the endometrial environment. ADAMTS1 is an interesting factor which is known as a secreted protease. It is involved in remodeling extracellular matrix [29]. Several studies showed that ADAMTS1 was detected in the uterus [11,30,31,32]. Ng and colleagues demonstrated that ADAMST1 was related to decidualization in human endometrium [32]. Kim and colleagues argued that ADAMTS1 expression in the uterus was maximum at metestrus stage [31].

Interestingly, YAP knockdown increased the expression of *AREG* and *AXL*. AREG is a transmembrane glycoprotein. Das et al. insisted that *AREG* is regulated by progesterone rather than estrogen and is important for embryonic implantation [33]. This case makes sense because estrogen and progesterone often induce opposing regulation. AXL, a receptor tyrosine kinase, is known to regulate various cellular processes such as cell survival, proliferation, and differentiation [34]. Although, there are fewer studies on AXL in the uterus, a recent study reported that AXL is highly detected in endometrial cancer [35].

The endometrium is a relatively simple tissue composed of luminal and glandular epithelial cells, and stromal cells, but it is a very delicate and precise tissue that changes dynamically with period. So far, the regulation of this dynamics has been studied from the perspective of two steroid hormones, such as estrogen and progesterone. However, there has always been a lack of explanation for these sophisticated modulations. We have studied the role of the Hippo signaling to better elucidate complex endometrial regulation with estrogen signaling [2]. Moreover, further studies related to relationship with progesterone will be important and interesting to understand the endometrial signaling regulation.

In conclusion, our current findings demonstrated that YAP, a key downstream target in the Hippo pathway, was regulated by estrogen and may have roles in the regulation of endometrial remodeling. These findings provide important insights into the complex signaling network of the endometrium during the estrus cycle.

## 4. Materials and Methods

### 4.1. Animal Management

All animal experiments were performed using adult (6- to 8-week-old) ICR mice provided by KOATECH (Pyeongtaek, Korea). Mice were housed under temperature- and light-controlled conditions with the lights on for 12 h daily and fed ad libitum. Care of mice and experimental and surgical procedures complied with the Guide for the Care and Use of Laboratory Animals and were approved by the Institutional Agricultural Animal Care and Use Committee of CHA University (approval No. IACUC190003).

### 4.2. Vaginal Smear Assay

The stages of the mouse estrous cycle were distinguished based on the morphology of epithelial cells in vaginal smear assays. Sterile phosphate-buffered saline (PBS) was gently shed into the vaginal vial using a pipette. PBS injected into the vaginal canal of 7-week-old ICR mice was pipetted several times and collected into 1.5-mL conical tubes. The collected PBS was dropped onto Superfrost Plus Stain slides (Thermo Fisher Scientific, Waltham, MA, USA), and the slides were dried on a 65 °C heat-block. Cell staining was performed using hematoxylin and eosin. After incubating the cells in 50, 75, and 90% ethanol for 5 min each, cells were stained with hematoxylin (Vector Laboratories, Burlingame, CA, USA) for 5 min and then washed in running water for 3 min. Samples were immersed in eosin Y (Sigma-Aldrich, St. Louis, MO, USA) for 10 min and then washed with 90 and 100% ethanol for 5 min each. After incubating cells in xylene, the slides were covered with coverslips using Permount Mounting Medium (Thermo Fisher Scientific, Waltham, MA, USA). After microscopic examination of the stained slides, the stages of the estrous cycle were determined based on the cytologic features described previously [18,36]. The proestrus stage showed an equal number of both leukocytes and nucleated epithelial cells. The estrus stage showed 75% nucleated epithelial cells and 25% non-nucleated epithelial cells. The metestrus stage showed an increased number of leukocytes with epithelial cells. The diestrus stage showed a predominance of leukocytes. When the estrous cycle was correctly identified, mice were placed in a CO_2_ chamber, and uterus samples were obtained. Half of the uterus was used for preparation of paraffin blocks, and the other half was used for RNA and protein isolation.

### 4.3. Ovariectomy and Hormone Treatments

To investigate the effects of estrogen on the expression of the *YAP* gene in the mouse uterus, 7-week-old ICR mice were subjected to ovariectomy (ovariectomized [OVX] mice) and then allowed to rest for 10 days before receiving hormone injections. The dorsal area of mice anesthetized using avertin (8–10 mg/mouse; Sigma-Aldrich, St. Louis, MO, USA) was shaved and swabbed with alcohol. An incision (approximately 1 cm) was made in the dorsal midline skin, and the ovary and oviduct were excised. The remaining tissue was replaced into the peritoneal cavity, and the incised skin was sutured. During recovery, OVX mice were examined for repair and infection for 10 days. After the recovery period, mice were injected with β-estradiol (E2, Sigma-Aldrich, St. Louis, MO, USA) at a concentration of 200 ng/mouse [18,37,38,39,40]. For assessment of responsiveness to hormones, mice were pre-injected with the estrogen receptor antagonist ICI 182,780 (ICI, Sigma-Aldrich, St. Louis, MO, USA; 500 μg/mouse) 30 min prior to hormone injection. 

### 4.4. Immunofluorescence

Rabbit polyclonal anti-YAP1 antibodies were purchased from Novus Biologicals (cat. no. NB110-58358, Centennial, CO, USA). Rabbit polyclonal anti-phospho-YAP (Ser127) antibodies (cat. No. 13008) from Cell Signaling Technology (Danvers, MA, USA) and YAP1 (phosphor-Ser127) antibody conjugated to FITC from Biorbyt Ltd. (Cambridge, UK) were used for the experiments. Uterus tissues harvested from mice were fixed in 4% paraformaldehyde for tissue immunostaining. After embedding in paraffin blocks, uterine tissues were sectioned to a thickness of 5 μm using a microtome (Macroteck, Goyang-si, Korea) and placed on a glass slide. For deparaffinization, slides were immersed in 100, 95, 70, and 50% ethanol for 5 min each and then washed in running water for 5 min. Deparaffinized slides were placed in a container with sodium citrate buffer (10 mM sodium citrate, 0.05% Tween 20, pH 6.0). The vessel was boiled for 40 min in an antigen retrieval steamer (IHC world, Gyeonggi-do, Korea). After cooling, the slides were washed once for 5 min in distilled water and three times for 5 min with 0.05% PBST buffer. A hydrophobic barrier was drawn around each tissue on the slide using an ImmEdge pen (Vector Labs, Burlingame, CA, USA). Slides were placed in a humidified chamber and then blocked with 5% goat serum blocking buffer (0.05% PBST with 4% bovine serum albumin). The chambers were incubated at room temperature for 2 h, treated with primary antibodies, and incubated at 4 °C overnight. Slides were washed three times for 5 min each in PBST and then treated with secondary Alexa-Fluor 488 or 546 antibodies (Thermo Fisher Scientific, Waltham, MA, USA) for 1 h at room temperature. Slides were then washed three times with PBS and treated with 4′,6-diamidino-2-phenylindole (DAPI; 1:10,000 dilution; Thermo Fisher Scientific, Waltham, MA, USA) for 10 min at room temperature. After washing the slides with PBS three times for 5 min each, a drop of mounting medium (DAKO, Glostrup, Denmark) was dropped on the slide and covered with a glass coverslip. Slides were observed with a confocal microscope LSM 880 (Carl Zeiss Co., Ltd., Oberkochen, Germany).

### 4.5. RNA Preparation

Total RNA was extracted from 6-week-old female mice using TRIzol (Invitrogen, Waltham, MA, USA) according to the manufacturer’s instructions. Potential genomic DNA was digested with RNase-free DNase (Qiagen, Hilden, Germany) during RNA purification. Complementary DNA (cDNA) was synthesized by reverse transcription of total RNA (2 μg) using a SensiFAST cDNA Synthesis Kit (Bioline, London, UK) according to the manufacturer’s instructions. 

### 4.6. Reverse Transcription Polymerase Chain Reaction (RT-PCR) and Real-Time PCR (qPCR)

RT-PCR was performed using a ProFlex PCR System and/or SimpliAmp Thermal Cycler (Applied Biosystems, Foster City, CA, USA). The thermal cycling parameters were as follows: denaturing at 95 °C for 30 s, annealing at 60–65 °C for 30 s, and extension at 72 °C for 30 s. The products were loaded and analyzed on 2% agarose gels. RT-qPCR analysis was performed using a CFX96 Touch Real-Time PCR Detection System (Bio-Rad Laboratories, Hercules, CA, USA) and/or QuantStudio 1 (Applied Biosystems, Foster City, CA, USA). iQ SYBR Green Supermix (Bio-Rad Laboratories, Hercules, CA, USA) and/or PowerUp SYBR Green Master Mix (Thermo Fisher Scientific, Waltham, MA, USA) were used for amplification. Cycling conditions for analysis were as follows: denaturation at 95 °C for 15 s, annealing at 60–65 °C (depending on primer conditions) for 15 s, and extension at 72 °C for 20 s (50 cycles). The comparative CT (ΔΔCT) method was used for relative gene expression analysis [41]. Relative values of gene expression were normalized to the relative expression of *RPL7* and/or *GAPDH* as reference genes. The annealing temperatures for each of the primer sets are shown in Table 1.

### 4.7. Western Blotting

The antibodies used for immunoblotting are as described above. Mouse monoclonal anti-β-actin antibodies (C-4) were purchased from Santa Cruz Biotechnology (Dallas, TX, USA). Boiled protein samples (30–80 μg) were loaded onto sodium dodecyl sulfate polyacrylamide gel electrophoresis gels (10% running gels) and electrophoresed at 80 V for 20 min and 100 V for 70 min. Using a semitransfer method, protein-loaded gels were placed on polyvinylidene difluoride membranes and electrophoresed at 20 V for 70 min. The membranes were blocked with ProNA phospho-block solution (TransLab, Seoul, Korea) at room temperature for 1 h. Specific primary antibodies were diluted with the blocking solution and incubated overnight at 4 °C. After washing three times with 0.05% PBST for 5 min each, appropriate HRP-conjugated secondary antibodies (OriGene Technologies, Rockville, MD, USA) were diluted 1:10,000 in ProNA phospho-block solution (TransLab, Seoul, Korea) and incubated for 1 h at room temperature. Membranes were washed with 0.2% PBST five times for 3 min each, and signals were detected using enhanced chemiluminescence (ECL) reactions with a Western ECL FAMTO Kit (LPS solution, Daejeon-si, Korea) or Amersham ECL Prime Western Blotting Detection Reagents (SeouLin Bioscience, Seongnam-si, Korea). Relative expression was analyzed using a ChemiDoc XRS system (Bio-Rad Laboratories, Hercules, CA, USA).

### 4.8. Statistical Analysis

All values are reported as mean ± SEM. The results were analyzed using one-way ANOVA for statistical evaluation with Microsoft Excel software (Microsoft, Redmond, WA, USA).

## Figures and Tables

**Figure 1 ijms-23-09772-f001:**
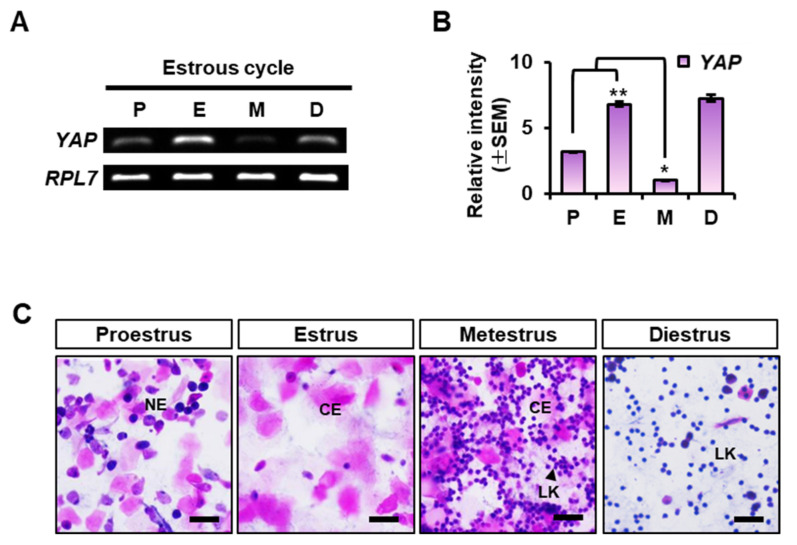
*YAP* expression in mouse uterus during the estrous cycle. (**A**) Relative expression of *YAP* mRNA in the mouse uterus at each stages during the estrous cycle by RT-PCR. Total RNA from the mouse uterus at each stages of the estrous cycle (P, proestrus; E, estrus; M, metestrus; D, diestrus) was isolated and amplified with specific primers for *YAP*. *RPL7* transcript was used as an internal control. (**B**) Quantification of the relative mRNA intensity of *YAP* in the mouse uterus using qRT-PCR. Expression levels were calculated using the ΔΔCT method and normalized to *RPL7* mRNA expression. * *p*-value < 0.01, ** *p*-value < 0.05. The fold changes were evaluated by comparing the level of *YAP* mRNA at the metestrus. (**C**) Representative images of vaginal smear assays confirming each stage of the estrous cycle. LK: leukocyte, NE: nucleated epithelial cells, CE: cornified epithelial cells. Scale bar: 100 μm.

**Figure 2 ijms-23-09772-f002:**
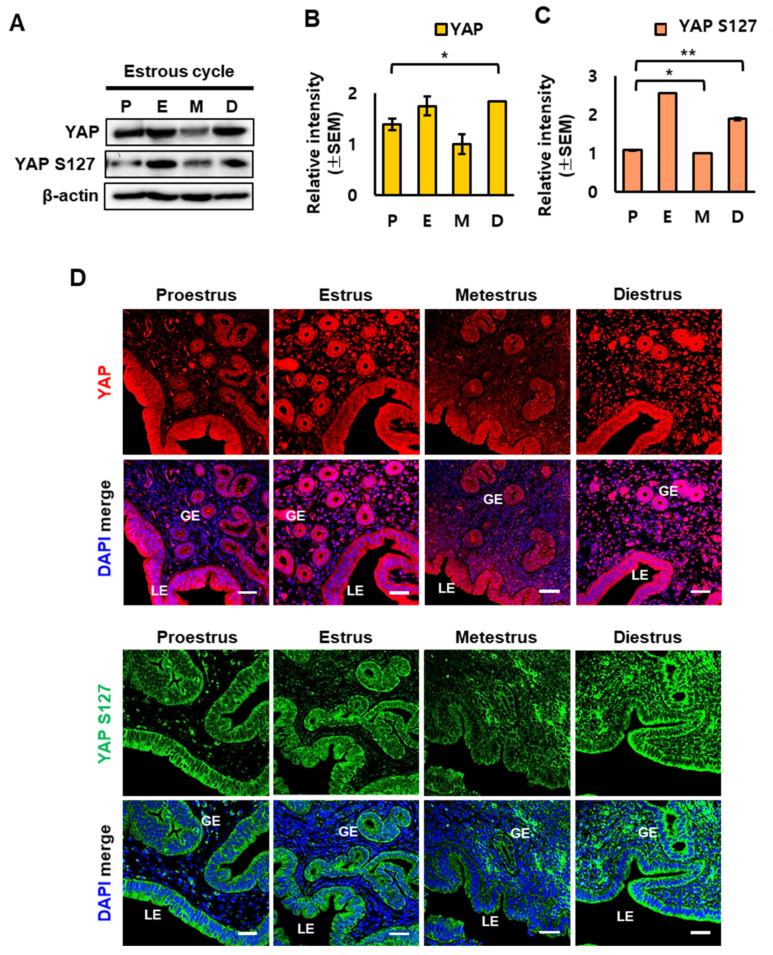
Localization of YAP and phospho-YAP in the mouse uterus during the estrous cycle. (**A**) Western blot analysis of YAP and phospho-YAP (YAP S127) proteins was performed using whole tissue lysates from the mouse uterus during the estrous cycle. (**B**,**C**) Quantification of relative protein levels of YAP and phospho-YAP (YAP S127) in the mouse uterus using ChemiDoc software. β-actin was used for normalization of protein level. * *p*-value < 0.1, ** *p*-value < 0.05. (**D**) Immunostaining for localization of YAP and phospho-YAP (YAP S127) proteins in the mouse uterus at each stage of the estrous cycle. DAPI was shown in blue. Scale bar: 25 µm. LE, luminal epithelium; GE, glandular epithelium.

**Figure 3 ijms-23-09772-f003:**
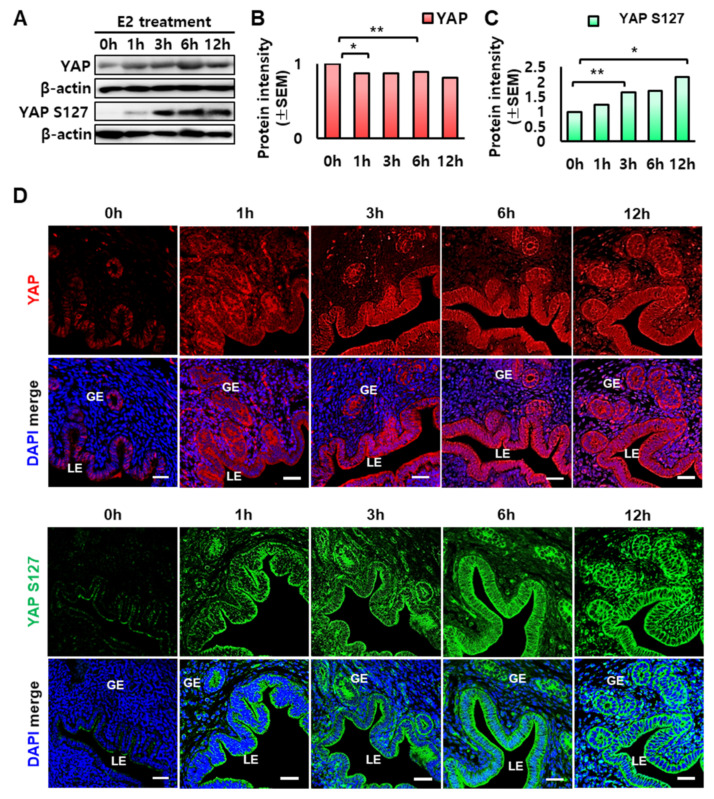
Effect of estrogen on YAP and phospho-YAP levels in the uterus of OVX mice. (**A**) Western blot analysis of YAP and phospho-YAP (YAP S127) proteins over time after estrogen treatment, 0, 1, 3, 6, 12 h (hour). (**B**,**C**) Relative quantification of western blot bands was performed using Image Lab software. * *p*-value < 0.01, ** *p*-value < 0.05. (**D**) Immunofluorescence for localization of YAP and phospho-YAP (YAP S127) proteins in the uterus of OVX mice treated with estrogen (E2; 200 ng/mouse). DAPI is shown in blue. LE: luminal epithelium, GE: glandular epithelium. Scale bar: 25 µm.

**Figure 4 ijms-23-09772-f004:**
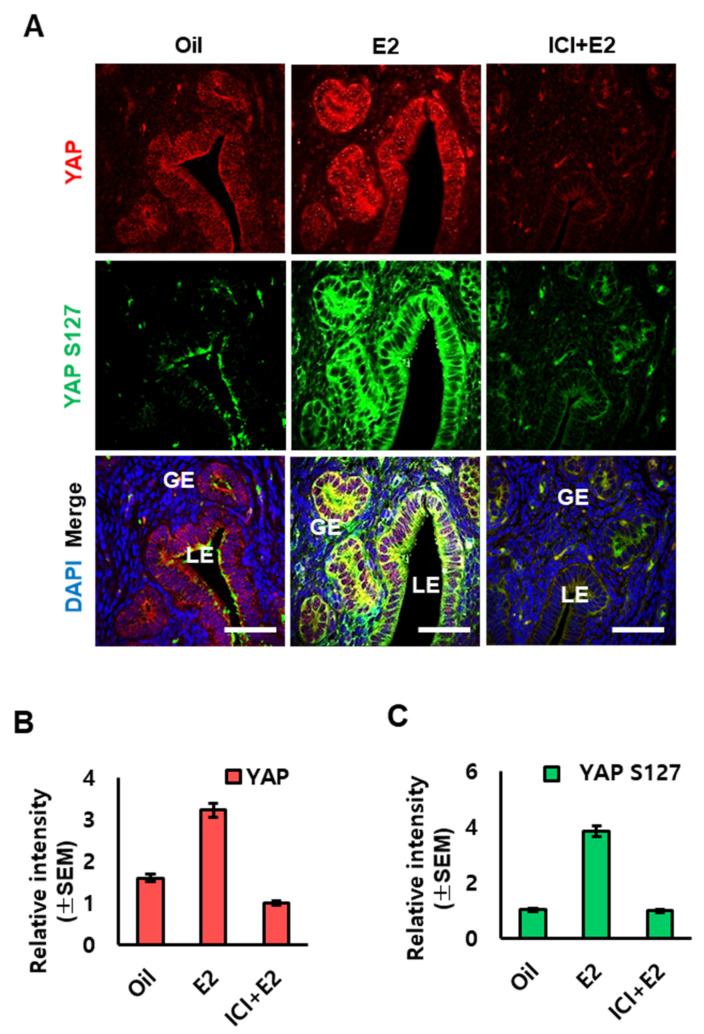
Effects of the estrogen receptor antagonist (ICI 182, 780, ICI) on the levels of YAP and phospho-YAP (YAP S127) in the uterus of OVX mice. (**A**) Cross-sections of uterus from sesame oil (oil; control), E2 only, and/or ICI+E2-treated OVX mice were stained with rabbit polyclonal antibodies against YAP or phospho-YAP (YAP S127). DAPI is shown in blue. LE, luminal epithelium; GE, glandular epithelium. Scale bar: 25 µm. (**B**,**C**) Relative quantitation of YAP and phospho-YAP (YAP S127) of immunofluorescence staining was analyzed using LAS AF Life software.

**Figure 5 ijms-23-09772-f005:**
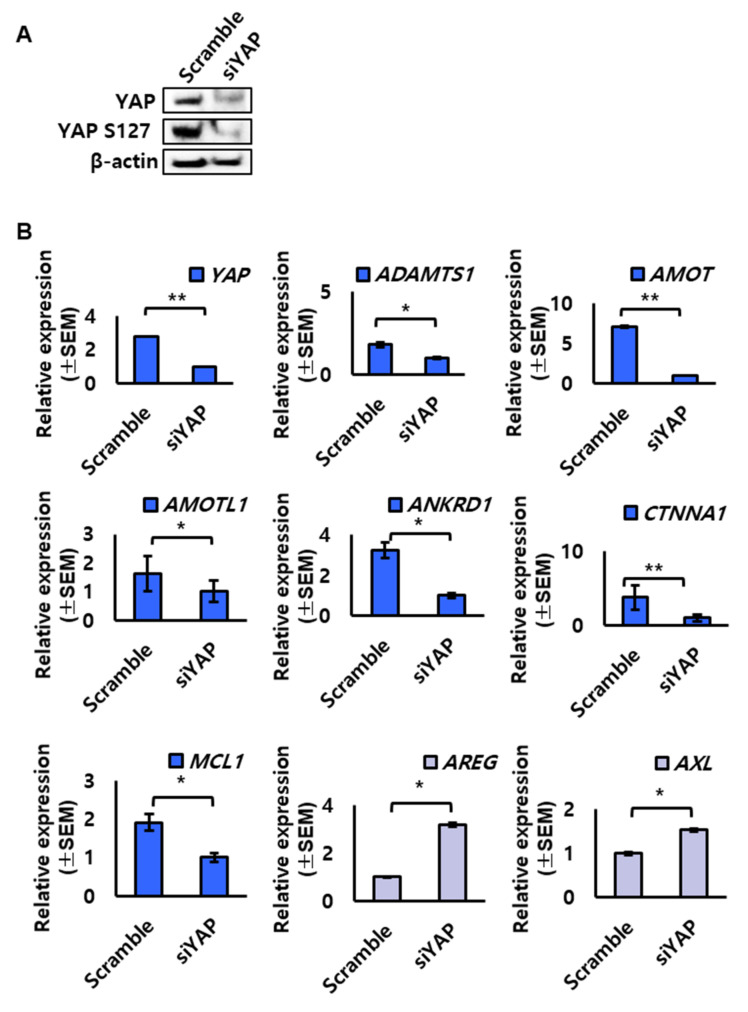
The expression of YAP target genes in endometrial cells with YAP knockdown. (**A**,**B**) *YAP* mRNA and protein were extracted from Ishikawa cells transfected with 20 μM of scrambled siRNA (scramble as a control) or *YAP* siRNA (siYAP). (**A**) Western blot analysis of YAP protein expression in Ishikawa cells. (**B**) qRT-PCR analysis for relative changes of YAP target gene expression upon *YAP* knockdown in Ishikawa cells. * *p*-value < 0.1, ** *p*-value < 0.05.

**Table 1 ijms-23-09772-t001:** Oligonucleotide sequences of primers used for RT-PCR and qRT-PCR.

Gene Symbol	NCBI ID	Sequence (5′–3′)	Annealing Temp.
*Human YAP*	NM_001195044.1	Fwd: CACAGCTCAGCATCTTCGAC	60 °C
		Rev: TATTCTGCTGCACTGGTGGA	
*Mouse YAP*	NM_001171147.1	Fwd: TCCAACCAGCAGCAGCAAAT	60 °C
		Rev: TTCCGTATTGCCTGCCGAAA	
*AMOT*	NM_001113490.2	Fwd: CATCACCACCAACAACAGCA	60 °C
		Rev: GTCTCCACCTTCTGCAGTCT	
*AMOTL1*	NM_001301007.2	Fwd: AGACAGACTCCTCCAGCCTA	60 °C
		Rev: CGTAGTGGAGGCTATGGAGG	
*AMOTL2*	NM_001363943.2	Fwd: GCTGCAGAGAGACAATGAGC	60 °C
		Rev: AGTCTTGCAGCCTCCTCATT	
*MCL1*	NM_021960.5	Fwd: GCTGCATCGAACCATTAGCA	60 °C
		Rev: ATGCCAAACCAGCTCCTACT	
*ANKRD1*	NM_014391.3	Fwd: TGAATCCACAGCCATCCACT	63 °C
		Rev: TCCTTCTCTGTCTTTGGCGT	
*AXL*	NM_001699.6	Fwd: GAGGGAGAGTTTGGAGCTGT	63 °C
		Rev: GAAACAGACACCGATGAGCC	
*ADAMTS1*	NM_006988.5	Fwd: GCAGAGCACTATGACACAGC	60 °C
		Rev: CACGTGGCCTAATTCATGGG	
*CTNNA1*	NM_001290307.3	Fwd: ATTAGTGGGGCTGCCTTGAT	60 °C
		Rev: GTCCCTGGTCTTCTTGGTCA	
*GAPDH*	AY340484.1	Fwd: ATGGGGAAGGTGAAGGTCG	60 °C
		Rev: ATTGTTGCCATCAATGACCC	
*RPL7*	NM_011291.5	Fwd: TTTGTCATCAGAATTCGAGG	60 °C
		Rev: CTGACTTCAGGTTGGGGTAC	

## Data Availability

Not applicable.

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
