# Peer review of "Estrogen Regulates the Expression and Localization of YAP in the Uterus of Mice"

_ijms, 2022, doi:10.3390/ijms23179772_

Round 1

Reviewer 1 Report

Thank you for the opportunity to review this article. This article is concise and only present the necessary data to prove the hypothesis. I have just one small comment about the Materials and Methods section description. I have some doubt, whether the description of routine standard techniques, such as H&E staining and detailed step by step description of IHC sample preparation, is really necessary in the main manuscript.

Otherwise the manuscript seems scientifically sound. Best of luck.

Author Response

Response to the 1st reviewer

Thank you for the opportunity to review this article. This article is concise and only present the necessary data to prove the hypothesis. I have just one small comment about the Materials and Methods section description. I have some doubt, whether the description of routine standard techniques, such as H&E staining and detailed step by step description of IHC sample preparation, is really necessary in the main manuscript.

Response: I fully understand. In fact, this manuscript was submitted to another journal for review, but was rejected. At that time, a reviewer requested a detailed description of the experimental method. So,we explained how to do it in detail.

Otherwise the manuscript seems scientifically sound. Best of luck.

Response: I appreciate it for the encouragement and comment.

Reviewer 2 Report

Minor editorial comments are as follows

1. Figure 1 A and B: Western blot of Figure 1A does not much with Figure 1B relative expression. Better to find another Westen blot Figure which matches Figure 1B.

2. line 202: there was always ben a lack of explanation? ben should be been

3. line 211: 6 to 8-week-old should be 6- to 8-week-old?

4. lines 224-225: delete  % ethanol after 50 and 70

5. line 228: delete % ethanol after 90

6. line 230: delete   , Waltham, MA, USA

7. lines 243, 248, 251: delete  , St. Louis, MO, USA

8. line 254: add (city, state, country) after 58358)

9. line 260: delete    % ethanol after 100, 95, 70

10. lines 272, 274: delete     , Waltham, MA, USA

11. line 292: delete     , Foster City, CA, USA

12. References: First letter of each word of journal names should be large capital: see Ref no. 1, 3, 4, 5, 6, 7, 9, 10, 11, 12, 13, 14, 15, 17, 20, 21, 22, 23, 24, 26, 27, 30, 31, 32, 33, 34, 36, 37, 38, 39, 40

13. References: titles of manuscript should be small capital, ref. no. 2, 7, 8, 16, 18, 25, 39

14. References: delete The in the journal names: Ref. no. 12, 13

15. line 389: delete : official journal of the International Association for the Study of the Liver

16. line 377: Drosophila should be italic?

12. line 293: delete    , Hercules, CA, USA

13. line 294: delete    , Waltham, MA, USA

14. line 319: delete   , Hercules, CA, USA

Author Response

We attached a file for the responses to reviewer 2.

Please find it for you.

Reviewer 3 Report

In this  study, the authors reported the regulation of Yap by estrogen/estrogen receptors in mouse uterus. Yap is highly expressed in the luminal epithelium and glandular epithelium. This work is of potential interest with good data.

I have several questions and comments as follows,

1)      Fig1C, it will be nice to have a brief introduction about how different cells indicate each stage of estrous cycle.

2)      Is there some change in the expression of estrogen receptors within the cycle of estrous?

3)      Again, Fig4, it’s helpful to check the possible change of ERs expression.

4)      Is Yap an ERs regulated gene? Some discussions of estrogen receptors- YAP expression- estrus will be nice.

5)      Do the authors have any hypotheses about another possible player progesterone, please discuss or comment.

6)      Please consider the title: Estrogen regulates the expression and localization of YAP in the uterus of estrous mice.

Author Response

We attached a file for the responses to reviewer 3.

Please find it for you.

Round 2

Reviewer 3 Report

All my questions have been addressed.